# PS-NPs Induced Neurotoxic Effects in SHSY-5Y Cells via Autophagy Activation and Mitochondrial Dysfunction

**DOI:** 10.3390/brainsci12070952

**Published:** 2022-07-20

**Authors:** Qisheng Tang, Tianwen Li, Kezhu Chen, Xiangyang Deng, Quan Zhang, Hailiang Tang, Zhifeng Shi, Tongming Zhu, Jianhong Zhu

**Affiliations:** Department of Neurosurgery, Huashan Hospital, Shanghai Medical College, Fudan University, National Center for Neurological Disorders, National Key Laboratory for Medical Neurobiology, Institutes of Brain Science, Shanghai Key Laboratory of Brain Function and Regeneration, MOE Frontiers Center for Brain Science, 12 Wulumuqi Zhong Rd., Shanghai 200040, China; tangqisheng@fudan.edu.cn (Q.T.); litianwen@outlook.com (T.L.); chenkzhu@163.com (K.C.); dxy623400630@163.com (X.D.); zhangquan_neuron@163.com (Q.Z.); hltang@fudan.edu.cn (H.T.); natureszf@139.com (Z.S.)

**Keywords:** polystyrene nanoparticles, neurotoxicity, autophagy, mitochondrial oxidative stress

## Abstract

Polystyrene nanoparticles (PS-NPs) are organic pollutants that are widely detected in the environment and organisms, posing potential threats to both ecosystems and human health. PS-NPs have been proven to penetrate the blood–brain barrier and increase the incidence of neurodegenerative diseases. However, information relating to the pathogenic molecular mechanism is still unclear. This study investigated the neurotoxicity and regulatory mechanisms of PS-NPs in human neuroblastoma SHSY-5Y cells. The results show that PS-NPs caused obvious mitochondrial damages, as evidenced by inhibited cell proliferation, increased lactate dehydrogenase release, stimulated oxidative stress responses, elevated Ca^2+^ level and apoptosis, and reduced mitochondrial membrane potential and adenosine triphosphate levels. The increased release of cytochrome c and the overexpression of apoptosis-related proteins apoptotic protease activating factor-1 (Apaf-1), cysteinyl aspartate specific proteinase-3 (caspase-3), and caspase-9 indicate the activation of the mitochondrial apoptosis pathway. In addition, the upregulation of autophagy markers light chain 3-II (LC3-II), Beclin-1, and autophagy-related protein (Atg) 5/12/16L suggests that PS-NPs could promote autophagy in SHSY-5Y cells. The RNA interference of Beclin-1 confirms the regulatory role of autophagy in PS-NP-induced neurotoxicity. The administration of antioxidant N-acetylcysteine (NAC) significantly attenuated the cytotoxicity and autophagy activation induced by PS-NP exposure. Generally, PS-NPs could induce neurotoxicity in SHSY-5Y cells via autophagy activation and mitochondria dysfunction, which was modulated by mitochondrial oxidative stress. Mitochondrial damages caused by oxidative stress could potentially be involved in the pathological mechanisms for PS-NP-induced neurodegenerative diseases.

## 1. Introduction

Polystyrene (PS) plastics are widely used as decorative materials in optical instruments and chemical departments, or they are utilized as vehicles for drugs and other types of health devices. It is estimated that several million tons of PS plastics are produced annually on a global scale. Over 80% of PS plastics are not or cannot be effectively recycled, and they are eventually released into the environment, particularly the marine environment. Discarded PS plastics can be physically broken into small pieces (nano∼micro size) and then ingested by organisms; the plastics can be accumulated in the food chain due to their resistance to biodegradation [1,2]. Previous studies have found that PS nanoparticles (PS-NPs) (with a diameter < 100 nm) are globally dispersed in marine sediments, oceans, and marine organisms [3,4]. Moreover, PS-NPs have been detected in various types of human food sources, such as bottled water, tap water, fish, shrimps, beer, sugar, and sea salt, indicating a potential risk to human health [5,6]. PS-NPs may be even more hazardous than parent macroparticles due to their greater permeability across biological membranes [7]. PS-NPs are also the ideal vector for various toxic contaminants, including persistent organic pollutants, metals, pathogenic bacteria, and antibiotics [8,9,10]. Therefore, investigation into the potential adverse effects and molecular mechanism induced by PS-NPs is of great urgency.

The uptake of PS-NPs by organisms and the subsequent translocation to different biological tissues was first reported decades ago [11]. Studies have so far confirmed that PS-NP exposure has multiple adverse effects, such as inflammatory responses, hepatic stress, intestinal damage, reproduction disorders, nephrotoxicity, and oxidative stress stimulation [12,13,14,15]. Recent studies have found that PS-NPs can be detected in the brain tissue of *Caenorhabditis elegans*, zebrafish, and mice [16,17,18,19]. The accumulated PS-NPs in mouse brains resulted in neurobehavior alteration and cognitive impairment [20]. Penetration by PS-NPs through the blood–brain barrier (BBB) is mainly due to increased BBB permeability [21], which could potentially result from the reduced expression of tight junction proteins [22]. Mechanism research conducted by Barboza et al. indicated that the neurotoxicity induced by PS-NPs in European seabass was mediated by the induction of oxidative stress, the promotion of lipid oxidation, and the inhibition of acetylcholinesterase (AChE) activity [23]. In vitro studies have confirmed that exposure to PS-NPs (100 nm) could induce oxidative stress, increase apoptosis, and alter metabolic activity in mice primary neuronal cells [24,25]. The uptake of PS-NPs by mice hippocampal neuronal HT22 cells also induced oxidative stress and cell cycle arrest [26]. It has been documented that oxidative stress is an important mechanism for the incidence of neurodegenerative diseases following chronic long-term exposure to NPs [27,28].

The autophagy process is involved in clearing and recycling damaged proteins and cellular components [29]. It has been demonstrated that autophagy plays an essential role in the protection of neurons in the central nervous system (CNS) [30]. Considerable evidence suggests that autophagy dysregulation induced by environmental stressors is involved in the development of many neurodegenerative diseases [30,31]. A recent study proved the involvement of the autophagy pathway in embryonic toxicity, nephrotoxicity, physiological toxicity, and oxidative damage induced by NPs [32,33,34]. Exposure to 60 nm PS-NPs in human neuroblastoma SH-SY5Y cells also demonstrated autophagic activation, which could potentially be responsible for neural tube defects [34]. However, relatively few mechanistic studies on autophagy in potential neurotoxic effects induced by PS-NP exposure have been conducted.

A size-dependent induction by PS-NPs of autophagy initiation was observed recently in vitro in human umbilical vein endothelial cells (HUVECs) [35]. It was demonstrated that PS-NPs with a diameter of less than 100 nm could directly penetrate the cell membrane and aggregate in the cytoplasm [24,34,35]. In this study, 50 nm PS-NPs were chosen for treating human neuroblastoma SHSY-5Y cells, which is one of the most commonly used in vitro models in neurotoxicity studies [36]. Assays of cytotoxicity, autophagy activation, and mitochondrial activity were conducted. RNA interference and antioxidant N-Acetyl-L-cysteine (N-Acetylcysteine, NAC) were applied to confirm the role of oxidative stress and the autophagy process.

## 2. Materials and Methods

### 2.1. Materials

The 50 nm polystyrene nanoparticle (PS-NPs) beads (5% *w*/*v*) were purchased commercially from Janus New-Materials (Nanjing, China) (refer to the Appendix A for more detailed parameters). SHSY-5Y cells, the clonal subline of neuroblastoma SK-N-SH cell line, were obtained from American Type Culture Collection (ATCC, Rockville, MD, USA). The 3-(4,5)-dimethylthiahiazo(-z-y1)-3,5-di- phenytetrazo-liumromide (MTT), Rhodamine 123 (Rh 123), 2,7-dichlorodi- hydrofluorescein diacetate (DCFH-DA), tert-Butyl hydroperoxide (tBHP), and N-Acetylcysteine (NAC) were obtained from Sigma (St. Louis, MO, USA). The annexin V-fluoresceine isothiocyanate/propidium iodide (annexin V-FITC/PI) apoptosis detection kit was obtained from Sungene (Tianjin, China). Protein extraction and assay kit was purchased from Thermo (MA, USA). Fluo 3-AM was acquired from Dojindo (Kyushu, Japan). Lactate dehydrogenase (LDH) cytotoxicity assay kit and adenosine 5′-triphosphate (ATP) detection kit were obtained from Beyotime (Shanghai, China).

### 2.2. Cell Culture and PS-NP Treatment

The complete culture medium used for the SHSY-5Y cells was RPMI 1640 (Hyclone, Logan, UT, USA) containing 10% FBS (Hyclone, Logan, UT, USA), 100 U/mL penicillin, and 100 mg/L streptomycin. The SHSY-5Y cells were sub-cultured at a 10^4^ cells/mL density and maintained in an incubator at 37 °C with 5% CO_2_ and 100% relative humidity. Working solutions of PS-NPs were freshly diluted using the RPMI 1640 medium to 20, 50, 100, 200, and 500 mg/L. For the cytotoxicity assays, SHSY-5Y cells in the exponentially growing phase were treated with different concentrations of PS-NPs from 20 to 500 mg/L. The treated groups of 100 and 200 mg/L PS-NPs were reserved for the following mechanistic studies based on the cytotoxicity results. NAC is a sulfhydryl-containing antioxidant that can protect SHSY-5Y cells from peroxidative stress. For the NAC experiments, SHSY-5Y cells were pretreated with NAC (5 mM, 4 h) and exposed to 200 mg/L PS-NPs for 24 h. The control group cells were treated with the RPMI 1640 medium only.

### 2.3. Cytotoxicity Assays

SHSY-5Y cells seeded in a 96-well plate were treated with different concentrations of PS-NPs for 24 h. MTT (5 mg/mL) was then supplied, and cell viability was measured by absorbance value (570 nm) using a multifunctional microplate reader (Tecan, Männedorf, Switzerland). LDH activity was detected by the LDH assay kit with an absorbance value recorded at 490 nm. Following exposure to PS-NPs, the intracellular reactive oxygen species (ROS) level of SHSY-5Y cells seeded in a 6-well plate was measured by a fluorescent DCFH-DA assay, with excitation at 480 nm and emission at 525 nm. At the same time, the oxidative stress inducer tBHP (100 μM, 1 h) was applied as a positive control for the detection of oxidative stress [37]. The mitochondrial membrane potential (MMP, Δψm) was measured with a fluorescent probe Rh 123 (10 μM), with excitation at 507 nm and emission at 529 nm. The calcium ion (Ca^2+^) content was examined by a fluorescent probe Fluo-3AM (5 μM), with excitation at 506 nm and emission at 526 nm. The fluorescence intensity was then converted to a percentage and compared to the control group [18]. The ATP levels after exposure to PS-NPs were measured with an ATP assay kit (Beyotime, Shanghai, China) according to the instructions of the manufacturer. Cell apoptosis was assayed using flow cytometry with the annexin V-FITC/PI kit according to the instructions. Detailed information related to these biological assays is provided in the Appendix A.

### 2.4. RNA Interference (RNAi)

RNAi regulates gene expression through small interfering RNA (20–24 bp). In this study, sh-pBeclin-1 plasmid (Dharmacon, Lafayette, CO, USA) was transfected into SHSY-5Y cells using the Lipofectamine 3000 (Invitrogen, Carlsbad, CA, USA) to construct Beclin-1 RNAi (siBeclin-1) cells according to the instructions. At the same time, sh-pGIPZ plasmid (Dharmacon, Lafayette, CO, USA) was applied to construct negative control (NC) cells. NC and siBeclin-1 cells were treated with 200 mg/L of PS-NPs so that the cytotoxicity results could be verified.

### 2.5. Reverse Transcription Quantitative Polymerase Chain Reaction (RT-qPCR)

Following PS-NP treatment (100 and 200 mg/L, 12 h), the SHSY-5Y cells were collected and lysed using TRIZOL reagent to prepare the total RNA. The first strand of cDNA was then reverse-transcribed using a reverse transcription kit (TOYOBO, Osaka, Japan). The cDNA was used as a template for real-time quantitative PCR amplification with the SYBR Green PCR Master Mix (Toyobo, Osaka, Japan). Glyceraldehyde-3-phosphate dehydrogenase (Gapdh) was used as an internal reference for quantifying the expression of Beclin-1. No-template controls were included in each experiment. More information, including the reaction mixture, amplification procedure, and data analysis, can be found in the Appendix A.

### 2.6. Immunofluorescence Detection

After treatments of 100 and 200 mg/L PS-NPs for 24 h, the SHSY-5Y cells were rinsed using Hank’s buffer and fixed with methanol for 10–20 min. After being washed once more using Hank’s buffer, the cells were treated with Triton-X 100 (0.1%) for 10–20 min and BSA Hank’s solution (1%) for 5–15 min. Then, cells were incubated with the primary antibody of cytochrome c (Cyc-c) (Abcam Company, Cambridgeshire, UK) and FITC-labeled secondary antibody (Beyotime, Shanghai, China), according to a previous study [38]. The fluorescence was excited and representative images were recorded using a fluorescent microscope (Olympus, Tokyo, Japan).

### 2.7. Western Blotting

After the PS-NP treatment described previously, the total protein samples of the SHSY-5Y cells were prepared using cell lysis buffer (Thermo, Waltham, MA, USA). The protein samples were subjected to sodium dodecyl sulfate-polyacrylamide gel electrophoresis and transferred to polyvinylidene fluoride membranes. The membranes were then sequentially incubated with the primary and secondary antibodies. More detailed information is provided in the Appendix A. The antibodies that were used in this study included anti-light chain 3-II (LC3-II), anti-Beclin-1, anti-autophagy-related protein 12 (Atg12), anti-Atg5, anti-Atg16L, anti-Cyc-c, anti-apoptotic protease activating factor-1 (Apaf-1), anti-cysteinyl aspartate specific proteinase-3 (caspase-3), anti-caspase-9 (Cell Signaling, MA, USA), anti-GAPDH (Abcam, Cambridgeshire, UK), horseradish peroxidase (HRP)-conjugated secondary anti-mouse immunoglobulin G (IgG), and anti-rabbit IgG (Zhong Shan, China).

### 2.8. Statistical Analysis

Each experiment was independently repeated three times with at least triplicate replicates for each group. Data were expressed as mean ± standard deviation (SD) and analyzed by one-way analysis of variance (ANOVA). Two-way ANOVA was performed to test the interaction between the inhibitor and non-inhibitor groups. A *p*-value < 0.05 was considered to be statistically significant.

## 3. Results and Discussion

### 3.1. PS-NPs Induced Neurotoxicity in SHSY-5Y Cells

As an emerging organic pollutant, the potential adverse effects of plastics have attracted extensive attention, especially nanoplastics with diameters of less than 100 nm. In this study, cytotoxicity was evaluated with MTT colorimetric assay, and the viability of SHSY-5Y cells treated with PS-NPs is shown in Figure 1A. Compared to the control group, a dose-dependent inhibitory effect on cell viability was observed in cells exposed to various concentrations of PS-NPs. Low concentrations of PS-NPs (<100 mg/L) appeared to have no obvious inhibitory effects on cell viability, while exposures of 200 and 500 mg/L PS-NPs for 24 h reduced cell viability by 12.0% and 29.6%, respectively (Figure 1A). The destruction of the cell membrane results in the release of the LDH enzyme, which is widely used as a membrane integrity indicator. PS-NPs could significantly induce the release of LDH in SHSY-5Y cells. After exposure to 200 and 500 mg/L PS-NPs for 24 h, the concentrations of LDH in the culture medium increased by 39.6% and 74.3%, respectively (Figure 1B). Previous studies have also observed a significant decrease in cell viability after exposure to PS-NPs (33 nm) under comparatively high concentrations (>180 mg/L) [39], as well as obvious increases in LDH activity after PS-NP (55 nm) treatment at 250 mg/L [25].

### 3.2. PS-NPs Promoted Oxidative Stress in SHSY-5Y Cells

The byproducts of mitochondrial metabolism include the buildup of reactive oxygen species (ROS) and the concentration of calcium ion (Ca^2+^) [40]. Excessive production of ROS can cause oxidative damage, disrupt homeostasis, and significantly suppress cell viability. Oxidative stress has been confirmed by previous studies to be a prominent neurotoxic mechanism after exposure to exogenous toxicants [41,42]. The activation of oxidative stress is closely associated with the onset of neurodegenerative diseases [43]. As shown in Figure 1C, PS-NP treatment at concentrations of >100 mg/L could obviously increase intracellular ROS production. There were 1.8-, 2.6-, and 3.0-fold increases in the ROS levels in the 100, 200, or 500 mg/L PS-NP treatment groups, respectively. An earlier study discovered that PS-NPs could significantly enter human cerebral microvascular endothelial hCMEC/D3 cells and mouse hippocampal neuronal HT22 cells, which resulted in a significant increase in ROS production [21,26].

### 3.3. PS-NPs Induced Mitochondrial Damage in SHSY-5Y Cells

As a secondary messenger, calcium ions (Ca^2+^) are essential for maintaining physiological nerve conduction function. Mitochondria take part in the regulation of the intracellular Ca^2+^ levels, functioning as a local Ca^2+^ buffer pool. The entry of Ca^2+^ into mitochondria is driven by the mitochondrial inner membrane (ΔΨm) electrochemical gradient. Ca^2+^ accumulation and the consequent mitochondrial permeability transition pore (mPTP) opening can lead to the dissipation of ΔΨm and ultimately induce the cell death signaling pathways [44]. In this study, the fluorescent probes Fluo-3AM and Rh123 were used to detect the Ca^2+^ concentrations and MMP in SHSY-5Y cells. Our results show that the Ca^2+^ level was markedly increased and the mitochondrial membrane potential (MMP, ΔΨm) was significantly reduced after treatment with 100, 200, and 500 mg/L PS-NP (Figure 1D,E), which indicates the induction of mitochondrial damage. These results are in agreement with previous findings that 55 nm PS-NPs influenced the mitochondrial activity of neural cells at high concentrations (250 mg/L) [25]. Other particles, such as combustion-derived particles, could also affect intracellular calcium homeostasis, contributing to the development or aggravation of cardiovascular disease [45].

Adenosine triphosphate (ATP) participates in various physiological processes in organisms, functioning as an active energy molecule, which is produced in the folds of the inner membrane of mitochondria. When the mitochondria are damaged, the switching state of the mitochondrial membrane voltage-dependent anion channel (VDAC) is affected, thereby inhibiting the release of macromolecule anionic ATP [46]. Especially in the state of apoptosis or necrosis, a decrease in ATP levels generally occurs simultaneously with the loss of MMP [47]. In comparison to the control group, the ATP levels in the SHSY-5Y cells were reduced by 13.2%, and 17.3%, respectively, after treatment with 200 and 500 mg/L PS-NP (Figure 1F). This finding further demonstrates the mitochondrial damage in SHSY-5Y cells induced by PS-NPs. All these data are consistent with the results of cell viability, suggesting that oxidative stress and mitochondrial respiration defect might be involved in the cytotoxicity induced by PS-NPs.

### 3.4. PS-NPs Induced Mitochondrial Apoptosis in SHSY-5Y Cells

Apoptosis is a complicated process in multicellular organisms to remove abnormal cells and maintain cellular homeostasis, which is regulated by complex signaling pathways [48]. The imbalance of apoptotic homeostasis is closely associated with numerous diseases, such as cancer, autoimmune diseases, neurological disorders, and cardiovascular disorders [49]. Oxidative stress has been proven to be one of the primary reasons for the induction of apoptosis [50]. To investigate whether the mitochondrial apoptotic pathway is involved in the cytotoxicity of PS-NPs in SHSY-5Y cells, cell apoptosis was detected using the annexin V-FITC/PI kit. Our results show a positive correlation between the apoptosis ratio and PS-NP concentration, with significant apoptosis induction at doses of 100, 200, and 500 mg/L (Figure 2A). In mammal cells, mitochondrial damage can cause the release of cytochrome c (Cyc-c) into the cytoplasm, forming a complex with cysteinyl aspartate specific proteinase-9 (caspase-9) and apoptotic protease activating factor-1 (Apaf-1), which can activate caspase-3, and ultimately result in apoptosis [51]. The localization transition of Cyc-c in cells was monitored in this study using an immunofluorescence assay. Based on the results of cytotoxicity, moderate doses of 100 and 200 mg/L were used for subsequent mechanistic studies. As shown in Figure 2B, the aggregation of Cyc-c in the cytoplasm was observed, indicating a release of Cyc-c from the mitochondria into the cytoplasm. The results of Western blotting show that the protein expressions of caspase-3 increased to 123.6% and 178.5% in the control group, after treatment with 100 and 200 mg/L PS-NPs. Synchronously, the expressions of caspase-9 and Aparf-1 show similar increasing trends to 136.2% and 185.3%, and 121.3% and 159.4% (Figure 2C,D), respectively. These results provide evidence that PS-NPs could trigger oxidative stress as an early response, eventually resulting in apoptosis via the mitochondrial apoptotic pathway.

### 3.5. PS-NPs Activated Autophagy in SHSY-5Y Cells

Autophagy is a conserved phylogenetic process, playing a vital role in the elimination of aggregated proteins and damaged organelles [29]. A dysregulated autophagy process is related to the development of neurodegenerative diseases [52]. Autophagic initiation is a complex multi-step and crosstalk process that could be induced by an alteration in redox signaling [53]. Excessive ROS is a well-known inducer of autophagy initiation [54]. Our results show that PS-NPs have the ability to induce oxidative stress and mitochondria dysfunction. To investigate the autophagy status induced by PS-NPs in SHSY-5Y cells, the expression levels of LC3-II were measured using Western blotting. LC3 is a homolog of autophagy-related protein-8 (Atg8) in mammalian cells. LC3-I (located in the cytoplasm) is covalently connected with phosphatidylethanolamine and inserted into the bilayer membrane of autophagy to form LC3-II. The conversion of LC3-I to LC3-II has been used as a recognized indicator for the evaluation and determination of autophagy [29]. As depicted in Figure 3A,B, a significant increase in LC3-II expression was observed after PS-NP exposure (Figure 3A), suggesting autophagy initiation in SHSY-5Y cells treated with PS-NPs.

To confirm the formation of autophagy in the SHSY-5Y cells, the expressions of Beclin-1 and Atg5/12/16L1 were measured using Western blotting. Beclin-1 is a subunit of the phosphatidylinositol 3 kinase complex. The binding of Beclin-1 to autophagy precursors results in the initiation of autophagosome formation [55]. The triplet complex that is formed by the sequential conjugation of Atg12, Atg5, and Atg16L is indispensable for autophagosome formation [56]. Our results show that the levels of Beclin-1 were elevated in SHSY-5Y cells exposed to PS-NPs compared to the control (Figure 3A,B). The expressions of Beclin-1 increased to 132.6% and 178.3%, respectively, after exposure to 100 and 200 mg/L PS-NPs. The expressions of Atg12, Atg5, and Atg16L show similar increasing changes, with inductions of 32.1–113.0% (Figure 3A,B). These results further confirm the induction of autophagy by PS-NPs. Wang et al. reported that human kidney proximal tubular epithelial HK-2 cells had higher protein levels of LC3-II and Beclin-1 after exposure to PS-MPs (2 μm) [57]. The data of Nie et al. indicated the upregulation of Atg7, Atg5, and LC3-II protein expression and the fluorescence intensity of GFP-LC3 in SH-SY5Y cells after exposure to 60 nm PS-NPs [34]. Exposure to 100 nm PS-NPs could increase LC3 autophagic pathway activation in mouse embryonic fibroblasts (MEFs) [58].

### 3.6. Regulatory Role of Autophagy in PS-NP-Induced Neurotoxicity

To verify the involvement of autophagy in the neurotoxicity induced by PS-NPs in SHSY-5Y cells, autophagy-blocked RNAi was conducted. Sh-pBeclin-1 and sh-pGIPZ plasmids were transfected into SHSY-5Y cells as described in the method section, and then the gene levels and protein expressions of Beclin-1 were analyzed using RT-qPCR and Western blotting, respectively. The results show that gene expression decreased by 76.5%, and protein expression decreased by 58.4% in siBeclin-1 SHSY-5Y cells compared to the control, indicating that autophagy was effectively knocked down in the initial step (Figure 4A). As displayed in Figure 4B, the upregulation of LC3-II induced by PS-NPs was obviously suppressed compared to NC cells. Similarly, in a study on sevoflurane-induced neurotoxicity, a significant decrease in LC3-II expression was observed after the transfection of siRNA Beclin-1 in neonatal rat hippocampal cells [59]. At the same time, biological influences on SHSY-5Y cells (including LDH release, ROS level, MMP, and apoptosis rate) were significantly relieved in siBeclin-1 SHSY-5Y cells. As shown in Figure 4C, the activity of LDH after exposure to 200 mg/L PS-NPs decreased by 14.5% in the siBeclin-1 cell group, and the production of ROS and the apoptosis rate were reduced by over 30%. In addition, the depolarization of MMP was also alleviated in the siBeclin-1 group. Previous studies have also reported that the inhibition of autophagy by Beclin-1 siRNA in neonatal rat hippocampal cells reduced sevoflurane-induced apoptosis [59]. The downregulation of Beclin-1 attenuated viral-induced inflammation in HIV-1-infected microglial cells [60]. These results indicate that autophagy is involved in the regulation of neurotoxicity induced by PS-NPs, which is also in accordance with the study conducted by Yin et al., showing that microplastics triggered autophagy-dependent ferroptosis and apoptosis in chicken brains [22].

### 3.7. PS-NP-Induced Neurotoxicity Attenuated by Oxidative Antioxidant

Accumulating evidence indicates the central role of autophagy in the mammalian oxidative stress response. The dysregulation of redox homeostasis can activate autophagy to degrade or recycle intracellular damaged macromolecules and facilitate cellular adaptation [61]. The antioxidant NAC can antagonize arsenic-induced neurotoxicity through the suppression of oxidative stress in mouse oligodendrocyte precursor cells [62]. The addition of NAC can abolish H_2_O_2_-induced autophagy and inhibit mitochondrial ROS production [63]. To confirm the role of oxidative stress in PS-NP-induced autophagy initiation and neurotoxicity, NAC was applied in the present study. NAC pretreatment obviously reduced ROS generation induced by PS-NPs in SHSY-5Y cells (Figure 5A). Similarly, LDH release, MMP reduction, and apoptosis induction in PS-NP-treated SHSY-5Y cells were significantly attenuated by the administration of NAC (Figure 5A). Additionally, NAC alleviated mitochondrial apoptosis in SHSY-5Y cells, demonstrated by reduced caspase-3, caspase-9, and Aparf-1 protein levels (Figure 5B). Moreover, the expressions of LC3-II and Beclin-1 expression in SHSY-5Y cells exposed to PS-NPs were significantly reduced by NAC pretreatment (Figure 5B). Liu et al. reported that the suppression of oxidative stress by NAC rescued the inhibition of cell growth and induction of autophagy caused by plasticizer tri-ortho-cresyl phosphate (TOCP) treatment in rat spermatogonia stem cells [64]. The pretreatment of antioxidant nanomaterial poly-amidoamine effectively impaired the autophagic effects and reduced neuronal cell death [65]. Our data confirm that mitochondrial oxidative stress plays a vital role in neurotoxicity and autophagy initiation induced by PS-NPs in SHSY-5Y cells, indicating that mitochondrial oxidative stress is a regulatory mechanism of neurotoxicity.

## 4. Conclusions

In conclusion, exposure to PS-NPs reduced cell viability and induced the release of LDH in SHSY-5Y cells in a concentration-dependent manner. PS-NPs could significantly induce oxidative stress and mitochondrial dysfunction as indicated by increased intracellular ROS production and Ca^2+^ concentrations and decreased MMP and ATP levels. Moreover, PS-NPs activated the mitochondrial apoptotic pathway and promoted cell apoptosis. The upregulation of autophagy marker proteins indicates the stimulation of the autophagy process after PS-NP exposure. The suppression of Beclin-1 with RNAi revealed the regulatory role of autophagy in neurotoxicity induced by PS-NPs. The results of antioxidant NAC suggest that mitochondrial oxidative stress could regulate PS-NP-induced neurotoxicity by the modulation of autophagy activation. In general, high concentrations (>100 mg/L) of PS-NPs resulted in significant neurotoxic effects by the activation of autophagy and mitochondrial dysfunction, which was modulated by oxidative stress. The stimulation of mitochondrial oxidative stress and autophagy suggests the potential pathological mechanisms of neurodegenerative diseases induced by PS-NPs.

## Figures and Tables

**Figure 1 brainsci-12-00952-f001:**
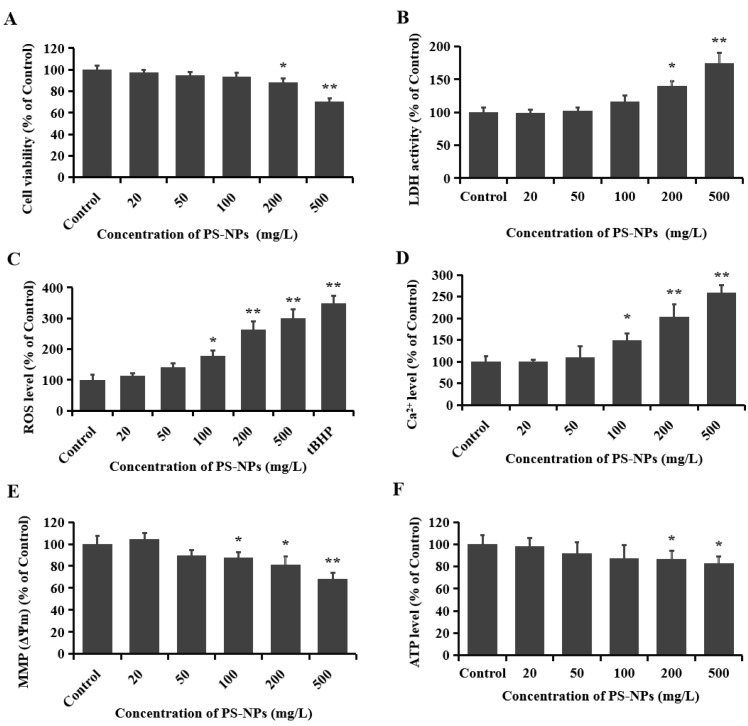
The cytotoxicity induced by PS-NPs in SHSY-5Y cells. SHSY-5Y cells were treated with different concentrations (20, 50, 100, 200, and 500 mg/L) of PS-NPs for 24 h. Then, cytotoxicity was measured. (**A**): Cell viability was measured with MTT. (**B**): LDH activity was detected using a LDH assay kit. (**C**): ROS production was measured using a fluorescent probe DCFH-DA, and tBHP (100 μM, 1 h) was used as a positive control for oxidative stress. (**D**): Ca^2+^ content was examined using the fluorescent probe Fluo-3AM (5 μM). (**E**): MMP (Δψm) was measured using the fluorescent probe Rh 123 (10 μM). (**F**): ATP level was measured using an ATP assay kit. The fluorescence intensity was converted as a percentage compared with the control. In the control group, SHSY-5Y cells were only treated with RPMI 1640 medium. * *p* < 0.05 and ** *p* < 0.01, compared to the control.

**Figure 2 brainsci-12-00952-f002:**
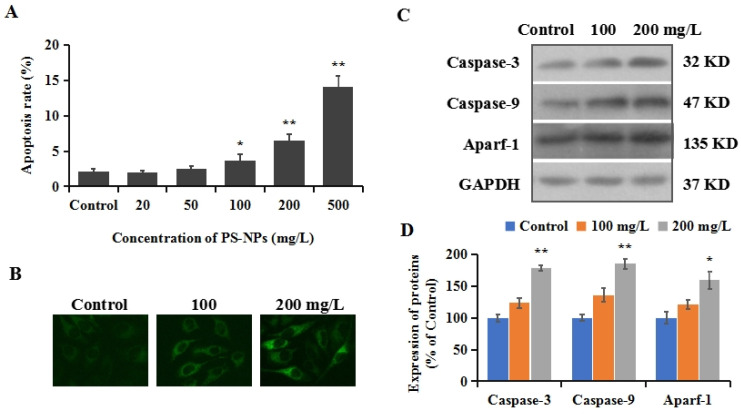
The apoptosis induced by PS-NPs in SHSY-5Y cells. (**A**): SHSY-5Y cells were treated with different concentrations (20, 50, 100, 200, and 500 mg/L) of PS-NPs for 24 h. Then, apoptosis was detected with the annexin V-FITC/PI kit. (**B**): After the treatment of 100 and 200 mg/L of PS-NPs for 24 h, the localization of Cyc-c was observed by immunofluorescence. (**C**): The expressions of caspase-3, caspase-9, and Aparf-1 were measured with Western blotting. (**D**): The expressions of proteins were quantified. In the control group, SHSY-5Y cells were only treated with medium. * *p* < 0.05 and ** *p* < 0.01, compared to the control.

**Figure 3 brainsci-12-00952-f003:**
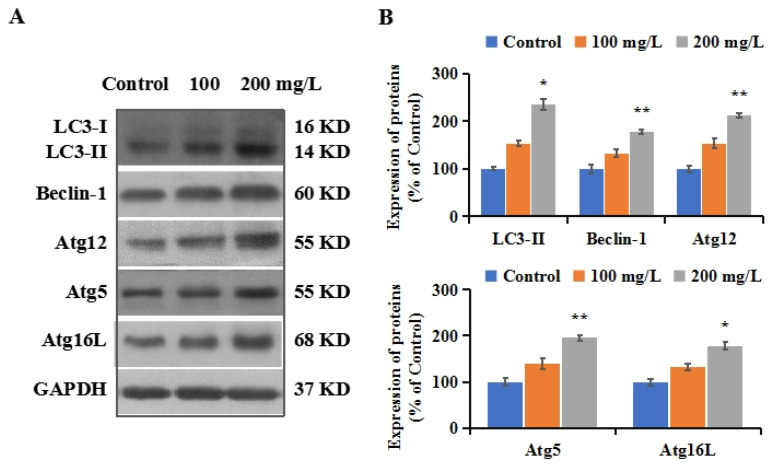
The autophagy induced by PS-NPs in SHSY-5Y cells. After the treatment of 100 and 200 mg/L of PS-NPs for 24 h. (**A**): The expressions of autophagy-related proteins including LC3-II, Beclin-1, Atg12, Atg5, and Atg16L were measured with Western blotting. (**B**): The expressions of proteins were quantified. In the control group, SHSY-5Y cells were only treated with medium. * *p* < 0.05 and ** *p* < 0.01, compared to the control.

**Figure 4 brainsci-12-00952-f004:**
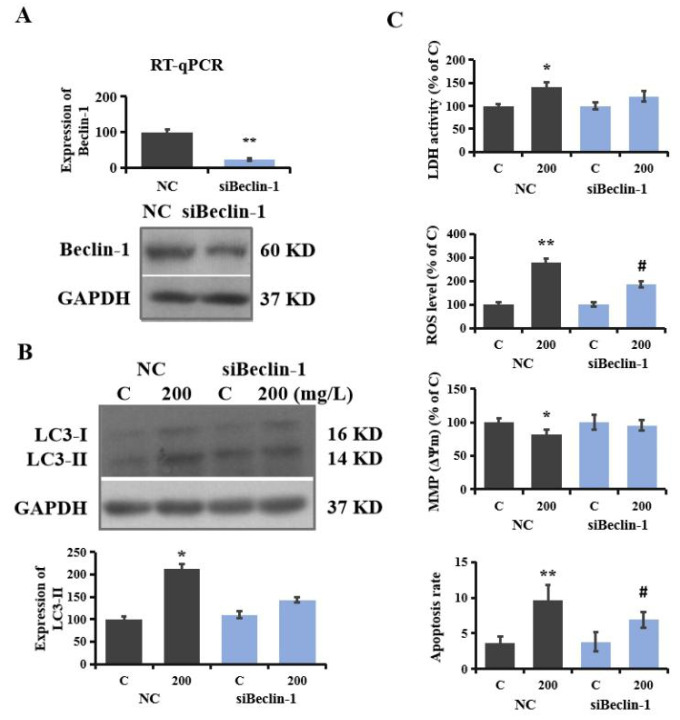
Autophagy-regulated neurotoxicity induced by PS-NPs. (**A**): SHSY-5Y cells were transfected with sh-pBeclin-1 and sh-pGIPZ plasmids to construct Beclin-1 RNAi cells (siBeclin-1) and negative cells (NC). The expression of Beclin-1 was analyzed with RT-qPCR and Western blotting. NC and siBeclin-1 cells were treated with 200 mg/L of PS-NPs, and then the expressions of LC3-II were measured with Western blotting, and the expressions of proteins were quantified (**B**). The LDH activity, ROS level, MMP, and apoptosis rate were measured (**C**). In the C (control) group, SHSY-5Y cells were only treated with medium. * *p* < 0.05 and ** *p* < 0.01, compared to the control. # *p* < 0.05, siBeclin-1 group compared to the NC group.

**Figure 5 brainsci-12-00952-f005:**
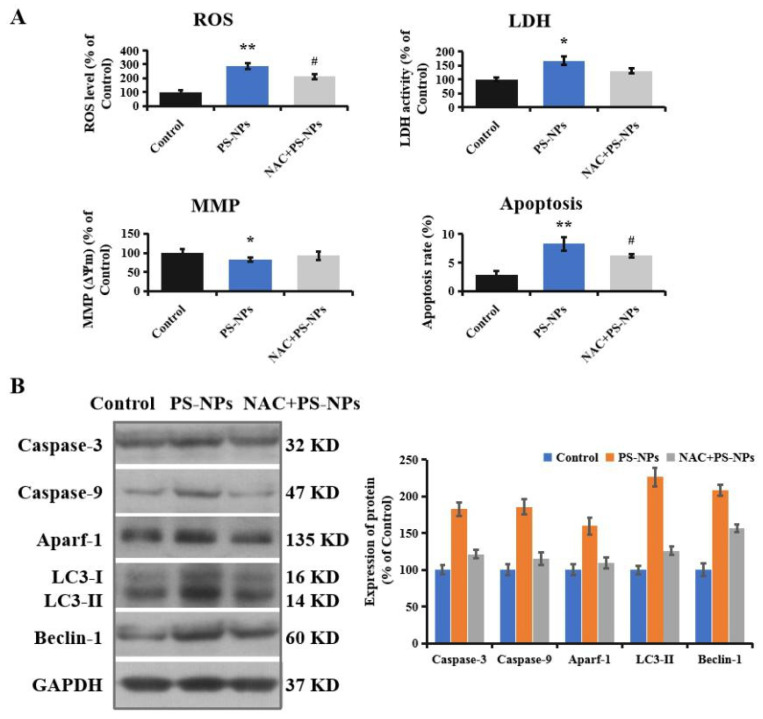
Oxidative stress regulated autophagy and neurotoxicity induced by PS-NPs. SHSY-5Y cells were pretreated with NAC (5 mM) for 4 h and then treated with 200 mg/L of PS-NPs for 24 h. (**A**): The ROS level, LDH activity, MMP, and apoptosis rate were measured. (**B**): The expression of apoptosis and autophagy-related proteins were measured with Western blotting, and the expressions of proteins were quantified. In the control group, SHSY-5Y cells were only treated with medium. * *p* < 0.05 and ** *p* < 0.01, compared to the control. # *p* < 0.05, NAC+PS-NPs group compared to the PS-NPs group.

## Data Availability

Data are contained within the article or Appendix A.

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
