# Peer review of "PS-NPs Induced Neurotoxic Effects in SHSY-5Y Cells via Autophagy Activation and Mitochondrial Dysfunction"

_brainsci, 2022, doi:10.3390/brainsci12070952_

Round 1
Reviewer 1 Report
Plastic pollution is a recognized environmental issue that can pose a health risk to humans. Specifically, fine particles resulting from the breakdown of plastics, such as microplastics and nanoplastics, are able to enter the food chain, accumulate in various tissues including the brain and even penetrate cell membranes interfering with cellular processes. The paper under review, titled “PS-Nps Induced Neurotoxic Effects in SHSY-5Y Cells via Autophagy Activation and Mitochondria Dysfunction”, examines the effect of polystyrene nanoparticles on the SHSY-5Y cell line and delineates the mode of cellular death behind the observed neurotoxicity. The paper’s main strength is that it employs a sound methodology, thoroughly investigating the effect of the nanoparticles on cellular viability utilizing a wide repertoire of methods. The authors examine major hallmarks of cell death alongside with changes in protein levels of mediators implicated in both apoptotic and autophagic procedures. Furthermore, apart from the observation they intervene by either attenuating the autophagy observed by Beclin-1 siRNA or alleviating the oxidative stress using n-acetylcysteine demonstrating their cell rescuing effect and effect on the chosen markers.
The paper’s major weakness is the use of English language and the style used. It is advised that both the paper and the supplemental materials edited by a native speaker. Minor weaknesses include the omission of some details in the methodology and result section that upon rectification could make the current work easier to reproduce and increase its transparency.
Specific comments
Line 26,27: “Above all”, “Meantime”. Those are some characteristic but not exhaustive examples where English language editing could benefit the readability of the study. Same holds true for the use of “and then” in supplementary material
Line 54: “of marine invertebrates (Caenorhabditis elegans)”. This might lead the reader to the erroneous assumption that C.elegans is a marine invertebrate whereas it is a terrestrial invertebrate. It should also be written in italics.
Line 105: “were sub cultured at a 104 cells/mL density”. It should probably read 104 cells/mL density
Line 139: While in Section 2.5 the internal reference (GAPDH) is mentioned, the target (Beclin-1) is omitted. It should appear on the main method as it appears in the supplementary material.
Line 167: Are the secondary antibodies HRP conjugated? It should be mentioned in the main text.
Line 171: What is the rationale behind using SEM over SD in the statistical analysis? It might be more informative for the reader to utilize SD in order to show the dispersion of individual experimental data points from the mean rather than SE.
Line 265,266 (Figure 2): To increase transparency Figure 2A should be optimally accompanied by flow cytometry dot plot showing the gates used. Same holds true for Figure 4C (Line 334-335). The small size of Figure 2B makes it difficult to interpret, it is also lacking information on magnification and a scale bar. If the concern is lack of space and picture layout, the requested material can be supplied in the supplementary material. Furthermore the Caspase-3 WB in the main paper and Caspase-3 WB in the supplementary material (Line 94) seem to have different orientations as far as up and down go. It can be rectified in order to avoid confusion.
Line 334,335 and 366,367 (Figure 4 and Figure 5): Figure 4C and Figure 5A employ a “#” symbol to denote the statistical significance of a comparison different than that of the asterisk (different control to which the sample of interest is compared to). While the figure caption clarifies to which comparison “#” refers to, in figure 4C the # symbol is situated on a line connecting the compared items, whereas in 5A the # symbol is put above the bar. I would suggest employing either a line with the symbol or the symbol directly above the bar but not both ways of depiction in order to keep the graphs consistent and uniform throughout the paper.
Also there seems to be a discrepancy between Figure 4A Beclin-1 WB in the main paper and the supplementary material (Line 97). I have attached a file where I highlight the differences with red circles in the two images. On the left is the image as it appears on the main document and on the right as it appears on the supplementary material.
Author Response
Response to reviewer
Thank you very much for your comments and suggestions, which are really helpful and greatly contribute to improve the quality of our MS. We have implemented all the comments in the revised manuscript. Below please find a detailed point-by-point response to each comment (in blue font).
Reviewer 1.
Plastic pollution is a recognized environmental issue that can pose a health risk to humans. Specifically, fine particles resulting from the breakdown of plastics, such as microplastics and nanoplastics, are able to enter the food chain, accumulate in various tissues including the brain and even penetrate cell membranes interfering with cellular processes. The paper under review, titled “PS-Nps Induced Neurotoxic Effects in SHSY-5Y Cells via Autophagy Activation and Mitochondria Dysfunction”, examines the effect of polystyrene nanoparticles on the SHSY-5Y cell line and delineates the mode of cellular death behind the observed neurotoxicity. The paper’s main strength is that it employs a sound methodology, thoroughly investigating the effect of the nanoparticles on cellular viability utilizing a wide repertoire of methods. The authors examine major hallmarks of cell death alongside with changes in protein levels of mediators implicated in both apoptotic and autophagic procedures. Furthermore, apart from the observation they intervene by either attenuating the autophagy observed by Beclin-1 siRNA or alleviating the oxidative stress using n-acetylcysteine demonstrating their cell rescuing effect and effect on the chosen markers.
The paper’s major weakness is the use of English language and the style used. It is advised that both the paper and the supplemental materials edited by a native speaker. Minor weaknesses include the omission of some details in the methodology and result section that upon rectification could make the current work easier to reproduce and increase its transparency.
Thank you very much for your advice. This paper and the supplemental material have been revised by native English speakers.
Specific comments
Line 26,27: “Above all”, “Meantime”. Those are some characteristic but not exhaustive examples where English language editing could benefit the readability of the study. Same holds true for the use of “and then” in supplementary material
Thank you very much for your comment. This paper and supplemental material have been revised by native English speakers, and some descriptions have been modified.
Line 54: “of marine invertebrates (Caenorhabditis elegans)”. This might lead the reader to the erroneous assumption that C.elegans is a marine invertebrate whereas it is a terrestrial invertebrate. It should also be written in italics.
Thank you very much for your reminding. The previous description is easy to misunderstand. “of marine invertebrates (Caenorhabditis elegans)” has been revised in MS. Please see: Line 56.
Line 105: “were sub cultured at a 104 cells/mL density”. It should probably read 104 cells/mL density
Thanks.The “104 cells/mL density” has been changed to “104 cells/mL density”. Please see: Line 108.
Line 139: While in Section 2.5 the internal reference (GAPDH) is mentioned, the target (Beclin-1) is omitted. It should appear on the main method as it appears in the supplementary material.
Thanks a lot for your suggestion. Glyceraldehyde-3-phosphate dehydrogenase (Gapdh) was used as an internal reference for quantifying the expression of Beclin-1. Please see: Line 149-151.
Line 167: Are the secondary antibodies HRP conjugated? It should be mentioned in the main text.
Yes. The horseradish peroxidase (HRP) conjugated secondary antibodies were applied, which have been added in main paper and supplemental material. Please see: Line 173-174.
Line 171: What is the rationale behind using SEM over SD in the statistical analysis? It might be more informative for the reader to utilize SD in order to show the dispersion of individual experimental data points from the mean rather than SE.
Thank you very much for your advice. It is a conceptual error. Actually, we utilized standard deviation in the statistical analysis, which has been modified in the MS. Please see: Line 177-180.
Line 265,266 (Figure 2): To increase transparency Figure 2A should be optimally accompanied by flow cytometry dot plot showing the gates used. Same holds true for Figure 4C (Line 334-335). The small size of Figure 2B makes it difficult to interpret, it is also lacking information on magnification and a scale bar. If the concern is lack of space and picture layout, the requested material can be supplied in the supplementary material. Furthermore the Caspase-3 WB in the main paper and Caspase-3 WB in the supplementary material (Line 94) seem to have different orientations as far as up and down go. It can be rectified in order to avoid confusion.
Thank you very much for your advice.
The flow cytometry dot plot of Figure 2A and Figure 4C have been added in supplementary material (Figure S1 and Figure S3).
The large size of Figure 2B with scale bar have been added in supplementary material (Figure S2).
The Caspase-3 WB of Figure 2C in supplementary material has been modified (Figure S5).
Line 334,335 and 366,367 (Figure 4 and Figure 5): Figure 4C and Figure 5A employ a “#” symbol to denote the statistical significance of a comparison different than that of the asterisk (different control to which the sample of interest is compared to). While the figure caption clarifies to which comparison “#” refers to, in figure 4C the # symbol is situated on a line connecting the compared items, whereas in 5A the # symbol is put above the bar. I would suggest employing either a line with the symbol or the symbol directly above the bar but not both ways of depiction in order to keep the graphs consistent and uniform throughout the paper.
Thank you very much for your constructive comment. We have uniformed the format, # symbol is put directly above the bar (Figure 4C and Figure 5A).
Also there seems to be a discrepancy between Figure 4A Beclin-1 WB in the main paper and the supplementary material (Line 97). I have attached a file where I highlight the differences with red circles in the two images. On the left is the image as it appears on the main document and on the right as it appears on the supplementary material.
Thank you very much for your advice. During chemiluminescence experiment, different exposure times cause subtle differences in the bands image. We have replaced another image in supplementary material (Figure S6).
Reviewer 2 Report
This study aim to examine the neurotoxicity, regulatory mechanisms and the protective role of N-acetylcysteine after polystyrene-NPs (PS-NPs) in Human neuroblastoma SHSY-5Y cells. This study is interesting because Nanoplastics have spread widely throughout not only the oceans but also the atmosphere, and recently created great concern about human health relevant to ingestion and accumulation of the nanoparticles by aquatic organisms in the human food-chain.
The cytotoxicity of PS-NPs was determine using MTT and LDH assays and oxidative stress was determine using mitochondrial membrane potential, calcium ion content and ATP levels. In this work also, Molecular biology and Immunolabeling techniques were developed using RT-qPCR, RNA Interference, Western Blotting and Immunofluorescence Detection.
The results revealed that PS-NPs had a significant neurotoxic effect on SHSY-5Y cells at all concentrations used, activates autophagy and activates the mitochondrial apoptosis pathway. N-acetylcysteine significantly attenuated the cytotoxicity and autophagy activation after PS-NPs exposure
Major comments
1) This research was performed on SHSY-5Y cells, but the authors did not mention whether the cells were differentiated. To take into account SH-SY5Y cells when using in vitro studies include both adherent and floating cells, both types of which are viable. The ability for researchers to differentiate SH-SY5Y neuroblastoma cells into cells possessing a more mature, neuron-like phenotype has afforded numerous benefits in the field of neuroscience research.
2) A dose-response assay is required, both for N-acetylcysteine (NAC) and tert-Butyl hydroperoxide (tBHP), in Measurement of cell viability
- In the case of NAC, the cells were treated with only a single dose (5 mM for 4 h), ideally several doses should have been used to established a working dose and to know that this dose does not affect the cells. This methodology is described in the work of Martínez et al., 2019 where they determined in SH-SY5Y cells the 1mM dose of NAC.
Martínez et al., 2019. Oxidative stress and related gene expression effects of cyfluthrin in human neuroblastoma SH-SY5Y cells: Protective effect of melatonin. Environ. Res. 177, 108579. https://doi.org/10.1016/j.envres.2019.108579.
- In the case of tBHP, the cells were treated with only a single dose (100 μM, exposure time is not specified). I would like to know why the authors choose this dosage? Did they carry out any previous experiments?
- In the section of Materials and methods, the 50 nm polystyrene nanoparticles (PS-NPs) were purchased commercially from Janus New-Materials and as they describe detailed parameters are as follows: diameter, 50 nm; solid content, 5%, w/v; coefficient of variation (c.v), ≤5%. They performed the characterisation of polystyrene nanoplastics such as: morphology and size of PS-NPs by transmission electron microscopy, hydrodynamic size and zeta potential of PS-NPs in Milli-Q water by dynamic light scattering?
3) In the case of the PS-NPs working solutions (20, 50, 100, 200 and 500 mg/L), it would have been ideal to set the IC30 or IC50 value to choose a certain dose to continue with the following experiments.
4) Figures and captions of figures should be corrected.
- In figure 1, line 237 and 238 indicate that figure 1D corresponds to the MPP test, when in fact they indicate Calcium levels. The same happens in Figure 1E indicating that it corresponds to calcium levels when in fact it is MPP.
- In Figure 2, line 268-27. The legend corresponding to figures 2A, 2B, 2C and 2D must be uniform in order to be understood. For example: Figure 2. The apoptosis induced by PS-NPs in SHSY-5Y cells. (A): SHSY-5Y cells were treated with different concentrations (20, 50, 100, 200, and 500 mg/L) of PS-NPs for 24 h. (B): Then apoptosis was detected with annexin V- FITC kit. After treatment with 100 and 200 mg/L of PS-NPs for 24 h, the localization of Cyc-c was observed by immunofluorescence (B), (C): the expressions of Caspase 3, Caspase 9, Aparf-1 were measured with Western blotting (C)
Minor comments
- Complete the Introduction and Discussion with further bibliography. In the introduction they should focus on in vitro and in vivo studies, in some paragraphs they cite work in fish.
- The authors should be careful in the preparation of the paper, for example in the section of Cell Culture and PS-NPs Treatment
Line 103: The authors do not describe the medium used, only 1640 is seen, I assume it is RPMI.
Line 105: The authors describe that "104 cells/ml", in my opinion it should be 104 cells/ml.
Author Response
Response to reviewer
Thank you very much for your comments and suggestions, which are really helpful and greatly contribute to improve the quality of our MS. We have implemented all the comments in the revised manuscript. Below please find a detailed point-by-point response to each comment (in blue font).
Reviewer 2.
Comments and Suggestions for Authors
This study aim to examine the neurotoxicity, regulatory mechanisms and the protective role of N-acetylcysteine after polystyrene-NPs (PS-NPs) in Human neuroblastoma SHSY-5Y cells. This study is interesting because Nanoplastics have spread widely throughout not only the oceans but also the atmosphere, and recently created great concern about human health relevant to ingestion and accumulation of the nanoparticles by aquatic organisms in the human food-chain.
The cytotoxicity of PS-NPs was determine using MTT and LDH assays and oxidative stress was determine using mitochondrial membrane potential, calcium ion content and ATP levels. In this work also, Molecular biology and Immunolabeling techniques were developed using RT-qPCR, RNA Interference, Western Blotting and Immunofluorescence Detection.
The results revealed that PS-NPs had a significant neurotoxic effect on SHSY-5Y cells at all concentrations used, activates autophagy and activates the mitochondrial apoptosis pathway. N-acetylcysteine significantly attenuated the cytotoxicity and autophagy activation after PS-NPs exposure
Major comments
1) This research was performed on SHSY-5Y cells, but the authors did not mention whether the cells were differentiated. To take into account SH-SY5Y cells when using in vitro studies include both adherent and floating cells, both types of which are viable. The ability for researchers to differentiate SH-SY5Y neuroblastoma cells into cells possessing a more mature, neuron-like phenotype has afforded numerous benefits in the field of neuroscience research.
Yes. SHSY-5Y cells have the properties of stem cells and are widely used in the study of the in vitro pathogenesis of nervous system diseases. Differentiated SHSY-5Y cells are more closely resemble mature neurons, with distinct neuronal morphology and long, broadly branched neuronal processes. In this study, undifferentiated and adherent SHSY-5Y cells were used. Differentiated SHSY-5Y cells will be applied in our following study of nanoplastics toxicity.
2) A dose-response assay is required, both for N-acetylcysteine (NAC) and tert-Butyl hydroperoxide (tBHP), in Measurement of cell viability
- In the case of NAC, the cells were treated with only a single dose (5 mM for 4 h), ideally several doses should have been used to established a working dose and to know that this dose does not affect the cells. This methodology is described in the work of Martínez et al., 2019 where they determined in SH-SY5Y cells the 1mM dose of NAC.
Martínez et al., 2019. Oxidative stress and related gene expression effects of cyfluthrin in human neuroblastoma SH-SY5Y cells: Protective effect of melatonin. Environ. Res. 177, 108579. https://doi.org/10.1016/j.envres.2019.108579.
Thank you very much for your advice. NAC is a widely used antioxidant with diverse dose and exposure time in multiple previous studies. For example, 1mM for 24 h (Martínez et al., 2019), 3 mM for 1 h (Wang et al., 2018), 2 mM for 2 h (Urano et al., (2018), 0.3-2 mM for 24 h (Song et al., 2021), and 4 or 10 mM for 24 h (Okamoto et al., 2016).
In our study, the concentration of NAC (5 mM for 4 h) was based on our previous experiment, which was provided in supplementary material (Figure S4A).
Ref:
Wang HF, Wang ZQ, Ding Y, Piao MH, Feng CS, Chi GF, Luo YN, Ge PF. Endoplasmic reticulum stress regulates oxygen-glucose deprivation-induced parthanatos in human SH-SY5Y cells via improvement of intracellular ROS. CNS Neurosci Ther. 2018 Jan;24(1):29-38. doi: 10.1111/cns.12771.
Urano Y, Mori C, Fuji A, Konno K, Yamamoto T, Yashirogi S, Ando M, Saito Y, Noguchi N. 6-Hydroxydopamine induces secretion of PARK7/DJ-1 via autophagy-based unconventional secretory pathway. Autophagy. 2018;14(11):1943-1958. doi: 10.1080/15548627.2018.1493043.
Song WJ, Yun JH, Jeong MS, Kim KN, Shin T, Kim HC, Wie MB. Inhibitors of Lipoxygenase and Cyclooxygenase-2 Attenuate Trimethyltin-Induced Neurotoxicity through Regulating Oxidative Stress and Pro-Inflammatory Cytokines in Human Neuroblastoma SH-SY5Y Cells. Brain Sci. 2021 Aug 24;11(9):1116. doi: 10.3390/brainsci11091116.
Okamoto A, Tanaka M, Sumi C, Oku K, Kusunoki M, Nishi K, Matsuo Y, Takenaga K, Shingu K, Hirota K. The antioxidant N-acetyl cysteine suppresses lidocaine-induced intracellular reactive oxygen species production and cell death in neuronal SH-SY5Y cells. BMC Anesthesiol. 2016 Oct 24;16(1):104. doi: 10.1186/s12871-016-0273-3.
- In the case of tBHP, the cells were treated with only a single dose (100 μM, exposure time is not specified). I would like to know why the authors choose this dosage? Did they carry out any previous experiments?
Yes. The tBHP is a common peroxide, which can induce the production of hydroxyl radicals, cause lipid peroxidation, and stimulate massive production of ROS. In this study, tBHP was used as a positive control to test the stability of the oxidative stress detection assay. The concentration (100 µM) and exposure time (1 h) of tBHP was selected according to our previous experiment, which were supplied in supplementary material (Figure S4B).
- In the section of Materials and methods, the 50 nm polystyrene nanoparticles (PS-NPs) were purchased commercially from Janus New-Materials and as they describe detailed parameters are as follows: diameter, 50 nm; solid content, 5%, w/v; coefficient of variation (c.v), ≤5%. They performed the characterisation of polystyrene nanoplastics such as: morphology and size of PS-NPs by transmission electron microscopy, hydrodynamic size and zeta potential of PS-NPs in Milli-Q water by dynamic light scattering?
Thanks. According to the supplier’s information, commercially PS-NPs have good monodispersity, template agent, and versatility, while these information cannot be used for publication. Please refer to the following URL.
http://www.nanojanus.com/en/product.php?mod=detail&id=19&cid=2&parentid=0
3) In the case of the PS-NPs working solutions (20, 50, 100, 200 and 500 mg/L), it would have been ideal to set the IC30 or IC50 value to choose a certain dose to continue with the following experiments.
Thank you very much. According to the results of MTT, the toxic effect of PS-NPs was comparatively weak. The inhibition rate of cell viability did not reach 30% even at the highest concentration (500 mg/L), so 100 and 200 mg/L were chosen for the following experiments.
4) Figures and captions of figures should be corrected.
- In figure 1, line 237 and 238 indicate that figure 1D corresponds to the MPP test, when in fact they indicate Calcium levels. The same happens in Figure 1E indicating that it corresponds to calcium levels when in fact it is MPP.
Thanks. Please forgive our negligence. The modification has been made. Please see: Line 247-249.
- In Figure 2, line 268-27. The legend corresponding to figures 2A, 2B, 2C and 2D must be uniform in order to be understood. For example: Figure 2. The apoptosis induced by PS-NPs in SHSY-5Y cells. (A): SHSY-5Y cells were treated with different concentrations (20, 50, 100, 200, and 500 mg/L) of PS-NPs for 24 h. (B): Then apoptosis was detected with annexin V- FITC kit. After treatment with 100 and 200 mg/L of PS-NPs for 24 h, the localization of Cyc-c was observed by immunofluorescence (B), (C): the expressions of Caspase 3, Caspase 9, Aparf-1 were measured with Western blotting (C)
Thanks. We would like to apologize for our negligence. The modification has been made. Please see: Line 280-282.
Minor comments
- Complete the Introduction and Discussion with further bibliography. In the introduction they should focus on in vitro and in vivo studies, in some paragraphs they cite work in fish.
Thanks. We have supplemented several latest references in the Introduction and Discussion sections. Some inappropriate descriptions have been adjusted in the MS. Please see: Line 55-69.
- The authors should be careful in the preparation of the paper, for example in the section of Cell Culture and PS-NPs Treatment
Line 103: The authors do not describe the medium used, only 1640 is seen, I assume it is RPMI.
Thanks. The modification has been made in the revised MS. Please see: Line 106, 110.
Line 105: The authors describe that "104 cells/ml", in my opinion it should be 104 cells/ml.
Thanks. Please forgive our negligence. This modification has been made in the Material and Method section. Please see: Line 108.
Reviewer 3 Report
The manuscript is focused on a topic of high interest for ecological and animal health: the effects of nano plastics dispensed in tons in the environment. However, it is not clear in the abstract. For general readers, the manuscript could be concerning nanoparticles as vehicles for drugs or other kinds of healthy devices. The suggestion is to introduce this issue in the abstract. Experiments performed are simple, but the novelty of the work should be more explored and discussed in the Results and Discussion section.
Importantly, to calculate the LC3II formation, it should be measured with a ratio with the housekeeping protein (in this case, GAPDH), never with LC3I. Based on the literature and autophagy guidelines, it is known that LC3I is more unstable than LC3II, generally being degraded with regular lysis buffer. Please, correct the measurement in figures 3 and 4.
Figures 2, 3, and 4 (WB) should show the approximated KDa from the protein ladder.
Some minor comments:
Introduction, lines 55-56: It is unclear what the authors want to say with the followed phrase "Furthermore, in vivo tests results revealed that PS-NPs accumulated in brain tissues of marine invertebrates (Caenorhabditis elegans) and fish (including Japanese rice fish, 54 Crucian carp, and zebrafish), could cause neurobehavior alteration and neurodevelopmental toxicity [16-18]." Is those animals or humans?
Introduction, lines 64: "such as Alzheimer syndrome (AD), Parkinson’s disease (PD)". Why Alzheimer's syndrome?
Results and Discussion, lines 194-195: "Excessive production of ROS can cause oxidative damage, disrupt homeostasis, significantly reduce cell viability and in-195 duce apoptosis or cell death." Why do authors separate these concepts here? Cell viability, apoptosis, cell death... Please, review for a better understanding.
Author Response
Response to reviewer
Thank you very much for your comments and suggestions, which are really helpful and greatly contribute to improve the quality of our MS. We have implemented all the comments in the revised manuscript. Below please find a detailed point-by-point response to each comment (in blue font).
Reviewer 3
Comments and Suggestions for Authors
The manuscript is focused on a topic of high interest for ecological and animal health: the effects of nano plastics dispensed in tons in the environment. However, it is not clear in the abstract. For general readers, the manuscript could be concerning nanoparticles as vehicles for drugs or other kinds of healthy devices. The suggestion is to introduce this issue in the abstract. Experiments performed are simple, but the novelty of the work should be more explored and discussed in the Results and Discussion section.
Thank you very much for you comment. Polystyrene nanoparticles (PS-NPs) are organic pollutants widely detected in environment and organisms, posing potential threats to ecosystem and human health. Abstract, and Introduction section have been modified.
Importantly, to calculate the LC3II formation, it should be measured with a ratio with the housekeeping protein (in this case, GAPDH), never with LC3I. Based on the literature and autophagy guidelines, it is known that LC3I is more unstable than LC3II, generally being degraded with regular lysis buffer. Please, correct the measurement in figures 3 and 4.
Thank you very much for you advice. The expression of LC3 II compared with GAPDH was quantified, and Figure 3B, Figure 4B and Figure 5B have been modified.
Figures 2, 3, and 4 (WB) should show the approximated KDa from the protein ladder.
Thank you very much for you suggestion. The information of molecular weight for target proteins have been added in Figure 2C, 3A, 4A, 4B, and 5B.
Some minor comments:
Introduction, lines 55-56: It is unclear what the authors want to say with the followed phrase "Furthermore, in vivo tests results revealed that PS-NPs accumulated in brain tissues of marine invertebrates (Caenorhabditis elegans) and fish (including Japanese rice fish, 54 Crucian carp, and zebrafish), could cause neurobehavior alteration and neurodevelopmental toxicity [16-18]." Is those animals or humans?
Thank you very much. This descriptions is indeed a bit confusing. We have revised the word description and supplemented some contents in the Introduction. Please see: Line 55-69.
Introduction, lines 64: "such as Alzheimer syndrome (AD), Parkinson’s disease (PD)". Why Alzheimer's syndrome?
These descriptions are inaccurate, and we have revised in MS.
Results and Discussion, lines 194-195: "Excessive production of ROS can cause oxidative damage, disrupt homeostasis, significantly reduce cell viability and in-195 duce apoptosis or cell death." Why do authors separate these concepts here? Cell viability, apoptosis, cell death... Please, review for a better understanding.
Thanks. These descriptions are inappropriate and have been adjusted. Please see: Line 202-204.
Round 2
Reviewer 1 Report
It is apparent that the authors have made a commendable effort to improve the manuscript according to the points raised. However, while they addressed the majority of the points, the manuscript still needs English editing in order for it to reach the level of a high quality international publication. Following, is a list of some grammatical errors detected that is by no means exhaustive. In order to fast track the procedure professional editing service can be utilized, since scientifically the manuscript is sound.
Main manuscript
13-15: Sentence structure. It reads as if chronic exposure penetrates BBB, where it is NPs
39: "healthy devices".
57: "Researches". Multiple instances. throughout the text in plural whereas it should be used in singular .
65: "is mainly as a result".
71-72: Sentence structure.
76: "arresting".
230: "could significantly internalized".
...
388: Sentence structure in the section title.
-
A manuscript section seems to have been left as it is in the template.
455-457: Please fill in the section.
Supplementary material
MTT section: A "And then" was overlooked.
Flow cytometry diagrams: The vertical axis reads ECD-A. Please add a legend explaining what it means. Is it the propidium iodide channel?
Author Response
It is apparent that the authors have made a commendable effort to improve the manuscript according to the points raised. However, while they addressed the majority of the points, the manuscript still needs English editing in order for it to reach the level of a high quality international publication. Following, is a list of some grammatical errors detected that is by no means exhaustive. In order to fast track the procedure professional editing service can be utilized, since scientifically the manuscript is sound.
Main manuscript
13-15: Sentence structure. It reads as if chronic exposure penetrates BBB, where it is NPs
39: "healthy devices".
57: "Researches". Multiple instances. throughout the text in plural whereas it should be used in singular .
65: "is mainly as a result".
71-72: Sentence structure.
76: "arresting".
230: "could significantly internalized".
...
388: Sentence structure in the section title.
-
A manuscript section seems to have been left as it is in the template.
455-457: Please fill in the section.
Supplementary material
MTT section: A "And then" was overlooked.
Thank you very much for your comments and suggestions. We have utilized the procedure professional editing service as recommended (MDPI) to for language, grammar and spell checking and professional editing. We hope the revised manuscript can meet the quality request of international publication. The grammatical errors listed in comments, as well as other language mistakes in MS, have been carefully checked and revised.
Flow cytometry diagrams: The vertical axis reads ECD-A. Please add a legend explaining what it means. Is it the propidium iodide channel?
Yes. The vertical axis is the propidium iodide channel, and the horizontal axis is the annexin V channel. We have added the explanation in the figure legend, and the images has been modified (Figure S1 and S3).
Reviewer 2 Report
Recommend acceptance. The manuscript has been significantly revised.
Author Response
Thank you very much.
Reviewer 3 Report
The revised manuscript is clearer, showing the relevance of the study for the environment and neurodegeneration fields. Unfortunately, the graphs from LC3II are still organized under the LC3I ratio, although the authors commented in their letter it was changed to a ratio regarded to the housekeeping protein (GAPDH). I am confused. Please, correct them in all graphs appearing LC3II measurement.
Author Response
The revised manuscript is clearer, showing the relevance of the study for the environment and neurodegeneration fields. Unfortunately, the graphs from LC3II are still organized under the LC3I ratio, although the authors commented in their letter it was changed to a ratio regarded to the housekeeping protein (GAPDH). I am confused. Please, correct them in all graphs appearing LC3II measurement
Thanks. Please forgive our negligence. The expression of LC3-II was measured compared with GAPDH, not compared with LC3-I. Practically, the anti-LC3 antibody used in this study was blotted with both LC3-II and LC3-I. The Western blotting pictures showed two bands, referring to LC3-II and LC3-I. The previous graphs was really misleading. The graphs of Figure 3A, Figure 4B and Figure 5B have been modified and re-organized. Thank you again for your advice.